# On the Expressive Power of
# Deep Polynomial Neural Networks

**Joe Kileel**[*]
Princeton University

**Matthew Trager**[*]
New York University

**Joan Bruna**
New York University

## Abstract

We study deep neural networks with polynomial activations, particularly *their expressive power*. For a fixed architecture and activation degree, a polynomial neural network defines an algebraic map from weights to polynomials. The image of this map is the functional space associated to the network, and it is an irreducible algebraic variety upon taking closure. This paper proposes *the dimension of this variety* as a precise measure of the expressive power of polynomial neural networks. We obtain several theoretical results regarding this dimension as a function of architecture, including an exact formula for high activation degrees, as well as upper and lower bounds on layer widths in order for deep polynomials networks to fill the ambient functional space. We also present computational evidence that it is profitable in terms of expressiveness for layer widths to increase monotonically and then decrease monotonically. Finally, we link our study to favorable optimization properties when training weights, and we draw intriguing connections with tensor and polynomial decompositions.

## 1 Introduction

A fundamental problem in the theory of deep learning is to study the *functional space* of deep neural networks. A network can be modeled as a composition of elementary maps, however the family of all functions that can be obtained in this way is extremely complex. Many recent papers paint an accurate picture for the case of shallow networks (*e.g.*, using mean field theory [7, 27]) and of deep linear networks [2, 3, 21], however a similar investigation of *deep nonlinear* networks appears to be significantly more challenging, and require very different tools.

In this paper, we consider a general model for *deep polynomial neural networks*, where the activation function is a polynomial ($r$-th power) exponentiation. The advantage of this framework is that the functional space associated with a network architecture is *algebraic*, so we can use tools from *algebraic geometry* [17] for a precise investigation of deep neural networks. Indeed, for a fixed activation degree $r$ and architecture $\boldsymbol{d} = (d_0, \dots, d_h)$ (expressed as a sequence of widths), the family of all networks with varying weights can be identified with an *algebraic variety* $\mathcal{V}_{\boldsymbol{d},r}$, embedded in a finite-dimensional Euclidean space. In this setting, an algebraic variety can be thought of as a manifold that may have singularities.

In this paper, our main object of study is the *dimension* of $\mathcal{V}_{\boldsymbol{d},r}$ as a variety (in practice, as a manifold), which may be regarded as a precise measure of the architecture's expressiveness. Specifically, we prove that this dimension stabilizes when activations are high degree, and we provide an exact dimension formula for this case (Theorem 14). We also investigate conditions under which $\mathcal{V}_{\boldsymbol{d},r}$ *fills* its ambient space. This question is important from the vantage point of optimization, since an architecture is "filling" if and only if it corresponds to a convex functional space (Proposition 6). In

---

[*]Equal contribution.

this direction, we prove a *bottleneck property*, that if a width is not sufficiently large, the network can never fill the ambient space regardless of the size of other layers (Theorem 19).

In a broader sense, our work introduces a powerful language and suite of mathematical tools for studying the geometry of network architectures. Although this setting requires polynomial activations, it may be used as a testing ground for more general situations and, *e.g.*, to verify rules of thumb rigorously. Finally, our results show that polynomial neural networks are intimately related to the theory of *tensor decompositions* [22]. In fact, representing a polynomial as a deep network corresponds to a type of decomposition of tensors which may be viewed as a composition of decompositions of a recently introduced sort [24]. Using this connection, we establish general non-trivial upper bounds on filling widths (Theorem 10). We believe that our work can serve as a first step towards many interesting research challenges in developing the theoretical underpinnings of deep learning.

## 1.1  Related work

The study of the expressive power of neural networks dates back to seminal work on the universality of networks as function approximators [10, 19]. More recently, there has been research supporting the hypothesis of "depth efficiency", *i.e.*, the fact that deep networks can approximate functions more efficiently than shallow networks [11, 25, 8, 9]. In contrast to this line of work, we study the class of functions that can be expressed *exactly* using a network. Our analysis may of course be used to investigate the problem of approximation, however this is not the focus of this paper.

Most of the aforementioned studies make strong hypotheses on the network architecture. In particular, [11, 25] focus on *arithmetic circuits*, or *sum-product networks* [29]. These are networks composed of units that compute either the product or a weighted sum of their inputs. In [8], the authors introduce a model of *convolutional arithmetic circuits*. This is a particular class of arithmetic circuits that includes networks with layers of 1D convolutions and product pooling. This model does not allow for non-linear activations (beside the product pooling), although the follow-up paper [9] extends some results to ReLU activations with sum pooling. Interestingly, these networks are related to Hierarchical Tucker (HT) decomposition of tensors.

The polynomial networks studied in this paper are not arithmetic circuits, but feedforward deep networks with polynomial $r$-th power activations. This is a vast generalization of a setting considered in several recent papers [33, 14, 31], that study shallow (two layer) networks with quadratic activations ($r = 2$). These papers show that if the width of the intermediate layer is at least twice the input dimension, then the quadratic loss has no "bad" local minima. This result in line with our Proposition 5, which explains in this case the functional space is convex and *fills* the ambient space. We also point out that polynomial activations are required for the functional space of the network to span a finite dimensional vector space [23, 33].

The polynomial networks considered in this paper do not correspond to HT tensor decompositions as in [8, 9], rather they are related to a different polynomial/tensor decomposition attracting very recent interest [16, 24]. These generalize usual decompositions, however their algorithmic and theoretical understanding are, mostly, wide open. Neural networks motivate several questions in this vein.

Finally, we mention other recent works that study neural networks from the perspective of algebraic geometry [26, 32, 20].

**Main contributions.**  Our main contributions can be summarized as follows.

- We give a precise formulation of the expressiveness of polynomial networks in terms of the algebraic dimension of the functional space as an *algebraic variety*.

- We spell out the close, two-way relationship between polynomial networks and a particular family of decompositions of tensors.

- We prove several theoretical results on the functional space of polynomial networks. Notably, we give a formula for the dimension that holds for sufficiently high activation degrees (Theorem 14) and we prove a tight lower bound on the width of the layers for the network to be "filling" in the functional space (Theorem 19).

**Notation.** We use $\mathrm{Sym}_d(\mathbb{R}^n)$ to denote the space of *homogeneous polynomials* of degree $d$ in $n$ variables with coefficients in $\mathbb{R}$. This set is a vector space over $\mathbb{R}$ of dimension $N_{d,n} = \binom{n+d-1}{d}$, spanned by all monomials of degree $d$ in $n$ variables. In practice, $\mathrm{Sym}_d(\mathbb{R}^n)$ is isomorphic to $\mathbb{R}^{N_{d,n}}$, and our networks will correspond to *points* in this high dimensional space. The notation $\mathrm{Sym}_d(\mathbb{R}^n)$ expresses the fact that a polynomial of degree $d$ in $n$ variables can always be identified with a *symmetric tensor* in $(\mathbb{R}^n)^{\otimes d}$ that collects all of its coefficients.

## 2 Basic setup

A *polynomial network* is a function $p_\theta : \mathbb{R}^{d_0} \to \mathbb{R}^{d_h}$ of the form

$$p_\theta(x) = W_h \rho_r W_{h-1} \rho_r \dots \rho_r W_1 x, \qquad W_i \in \mathbb{R}^{d_i \times d_{i-1}},$$

where the *activation* $\rho_r(z)$ raises all elements of $z$ to the $r$-th power ($r \in \mathbb{N}$). The parameters $\theta = (W_h, \dots, W_1) \in \mathbb{R}^{d_\theta}$ (with $d_\theta = \sum_{i=1}^h d_i d_{i-1}$) are the network's *weights*, and the network's *architecture* is encoded by the sequence $\boldsymbol{d} = (d_0, \dots, d_h)$ (specifying the *depth* $h$ and *widths* $d_i$). Clearly, $p_\theta$ is a homogeneous polynomial mapping $\mathbb{R}^{d_0} \to \mathbb{R}^{d_h}$ of degree $r^{h-1}$, *i.e.*, $p_\theta \in \mathrm{Sym}_{r^{h-1}}(\mathbb{R}^{d_0})^{d_h}$.

For fixed degree $r$ and architecture $\boldsymbol{d} = (d_0, \dots, d_h)$, there exists an algebraic map

$$\Phi_{\boldsymbol{d},r} : \theta \mapsto p_\theta = \begin{bmatrix} p_{\theta 1} \\ \vdots \\ p_{\theta d_{h+1}} \end{bmatrix}, \tag{1}$$

where each $p_{\theta i}$ is a polynomial in $d_0$ variables. The image of $\Phi_{\boldsymbol{d},r}$ is a set of vectors of polynomials, *i.e.*, a subset $\mathcal{F}_{\boldsymbol{d},r}$ of $\mathrm{Sym}_{r^{h-1}}(\mathbb{R}^{d_0})^{d_h}$, and it is the *functional space* represented by the network. In this paper, we consider the "Zariski closure" $\mathcal{V}_{\boldsymbol{d},r} = \overline{\mathcal{F}_{\boldsymbol{d},r}}$ of the functional space.[1] We refer to $\mathcal{V}_{\boldsymbol{d},r}$ as *functional variety* of the network architecture, as it is in fact an irreducible *algebraic variety*. In particular, $\mathcal{V}_{\boldsymbol{d},r}$ can be studied using powerful machinery from *algebraic geometry*.

**Remark 1.** *The functional variety $\mathcal{V}_{\boldsymbol{d},r}$ may be significantly larger than the actual functional space $\mathcal{F}_{\boldsymbol{d},r}$, since the Zariski closure is typically larger than the closure with respect to the standard the Euclidean topology. On the other hand, the* dimensions *of the spaces $\mathcal{V}_{\boldsymbol{d},r}$ and $\mathcal{F}_{\boldsymbol{d},r}$ agree, and the set $\mathcal{V}_{\boldsymbol{d},r}$ is usually "nicer" (it can be described by polynomial equations, whereas an exact implicit description of $\mathcal{F}_{\boldsymbol{d},r}$ may require inequalities).*

### 2.1 Examples

We present some examples that describe the functional variety $\mathcal{V}_{\boldsymbol{d},r}$ in simple cases.

**Example 2.** *A* linear network *is a polynomial network with $r = 1$. In this case, the network map $\Phi_{\boldsymbol{d},r} : \mathbb{R}^{d_\theta} \to \mathrm{Sym}_1(\mathbb{R}^{d_0})^{d_h} \cong \mathbb{R}^{d_h \times d_0}$ is simply matrix multiplication:*

$$\theta = (W_h, W_{h-1}, \dots, W_1) \mapsto p_\theta = W_h W_{h-1} \dots W_1 x.$$

*The functional space $\mathcal{F}_{\boldsymbol{d},r} \subseteq \mathbb{R}^{d_h \times d_0}$ is the set of matrices with rank at most $d_{\min} = \min_i \{d_i\}$. This set is already characterized by polynomial equations, as the common zero set of all $(1 + d_{\min}) \times (1 + d_{\min})$ minors, so $\mathcal{F}_{\boldsymbol{d},r} = \mathcal{V}_{\boldsymbol{d},r}$ in this case. The dimension of $\mathcal{V}_{\boldsymbol{d},r} \subset \mathbb{R}^{d_h \times d_0}$ is $d_{\min}(d_0 + d_h - d_{\min})$.*

**Example 3.** *Consider $\boldsymbol{d} = (2, 2, 3)$ and $r = 2$. The input variables are $x = [x_1, x_2]^T$, and the parameters $\theta$ are the weights*

$$W_1 = \begin{bmatrix} w_{111} & w_{112} \\ w_{121} & w_{122} \end{bmatrix}, \quad W_2 = \begin{bmatrix} w_{211} & w_{212} \\ w_{221} & w_{222} \\ w_{231} & w_{232} \end{bmatrix}.$$

*The network map $p_\theta$ is a triple of quadratic polynomials in $x_1, x_2$, that can be written as*

$$W_2 \rho_2 W_1 x = \begin{bmatrix} w_{211}(w_{111}x_1 + w_{112}x_2)^2 + w_{212}(w_{121}x_1 + w_{122}x_2)^2 \\ w_{221}(w_{111}x_1 + w_{112}x_2)^2 + w_{222}(w_{121}x_1 + w_{122}x_2)^2 \\ w_{231}(w_{111}x_1 + w_{112}x_2)^2 + w_{232}(w_{121}x_1 + w_{122}x_2)^2 \end{bmatrix}. \tag{2}$$

*The map $\Phi_{(2,2,3),2}$ in (1) takes $W_1, W_2$ (that have $d_\theta = 10$ parameters) to the three quadratics in $x_1, x_2$ displayed above. The quadratics have a total of $\dim \mathrm{Sym}_2(\mathbb{R}^2)^3 = 9$ coefficients, however these coefficients are not arbitrary, i.e., not all possible triples of polynomials occur in the functional space. Writing $c_{ij}^{(k)}$ for the coefficient of $x_i x_j$ in $p_{\theta k}$ in (2) (with $k = 1, 2, 3$ $i, j = 1, 2$) then it is a simple exercise to show that*

$$\det \begin{bmatrix} c_{11}^{(1)} & c_{12}^{(1)} & c_{22}^{(1)} \\ c_{11}^{(2)} & c_{12}^{(2)} & c_{22}^{(2)} \\ c_{11}^{(3)} & c_{12}^{(3)} & c_{22}^{(3)} \end{bmatrix} = 0.$$

*This cubic equation describes the functional variety $\mathcal{V}_{(2,3,3),2}$, which is in this case an eight-dimensional subset (hypersurface) of $\mathrm{Sym}_2(\mathbb{R}^2)^3 \cong \mathbb{R}^9$.*

## 2.2 Objectives

The main goal of this paper is to study the *dimension* of $\mathcal{V}_{\boldsymbol{d},r}$ as the network's architecture $\boldsymbol{d}$ and the activation degree $r$ vary. This dimension may be considered a *precise* and *intrinsic* measure of the polynomial network's *expressivity*, quantifying degrees of freedom of the functional space. For example, the dimension reflects the number of input/output pairs the network can interpolate, as each sample imposes one linear constraint on the variety $\mathcal{V}_{\boldsymbol{d},r}$.

In general, the variety $\mathcal{V}_{\boldsymbol{d},r}$ lives in the ambient space $\mathrm{Sym}_{r^{h-1}}(\mathbb{R}^{d_0})^{d_h}$, which in turn only depends on the activation degree $r$, network depth $h$, and the input/output dimensions $d_0$ and $d_h$. We are thus interested in the role of the intermediate widths in the dimension of $\mathcal{V}_{\boldsymbol{d},r}$.

**Definition 4.** *A network architecture $\boldsymbol{d} = (d_0, \ldots, d_h)$ has a* filling functional variety *for the activation degree $r$ if $\mathcal{V}_{\boldsymbol{d},r} = \mathrm{Sym}_{r^{h-1}}(\mathbb{R}^{d_0})^{d_h}$.*

It is important to note that if the functional variety $\mathcal{V}_{\boldsymbol{d},r}$ is filling, then actual functional space $\mathcal{F}_{\boldsymbol{d},r}$ (before taking closure) is in general only *thick*, i.e., it has positive Lebesgue measure in $\mathrm{Sym}_{r^{h-1}}(\mathbb{R}^{d_0})^{d_h}$ (see Remark 1). On the other hand, given an architecture with a thick functional space, we can find another architecture whose functional space is the whole ambient space.

**Proposition 5** (Filling functional space). *Fix $r$ and suppose $\boldsymbol{d} = (d_0, d_1, \ldots, d_{h-1}, d_h)$ has a filling functional variety $\mathcal{V}_{\boldsymbol{d},r}$. Then the architecture $\boldsymbol{d}' = (d_0, 2d_1, \ldots, 2d_{h-1}, d_h)$ has a filling functional space, i.e., $\mathcal{F}_{\boldsymbol{d}',r} = \mathrm{Sym}_{r^{h-1}}(\mathbb{R}^{d_0})^{d_h}$.*

In summary, while an architecture with a filling functional variety may not necessarily have a filling functional space, it is sufficient to double all the intermediate widths for this stronger condition to hold. As argued below, we expect architectures with thick/filling functional spaces to have more favorable properties in terms of optimization and training. On the other hand, non-filling architectures may lead to interesting functional spaces for capturing patterns in data. In fact, we show in Section 3.2 that non-filling architectures generalize families of low-rank tensors.

## 2.3 Connection to optimization

The following two results illustrate that thick/filling functional spaces are helpful for optimization.

**Proposition 6.** *If the closure of a set $C \subset \mathbb{R}^n$ is not convex, then there exists a convex function $f$ on $\mathbb{R}^n$ whose restriction to $C$ has arbitrarily "bad" local minima (that is, there exist local minima whose value is arbitrarily larger than that of a global minimum).*

**Proposition 7.** *If a functional space $\mathcal{F}_{\boldsymbol{d},r}$ is not thick, then it is not convex.*

These two facts show that if the functional space is not thick, we can always find a convex loss function and a data distribution that lead to a landscape with arbitrarily bad local minima. There is also an obvious weak converse, namely that if the functional space is filling $\mathcal{F}_{\boldsymbol{d},r} = \mathrm{Sym}_{r^{h-1}}(\mathbb{R}^{d_0})^{d_h}$, then any convex loss function $\mathcal{F}_{\boldsymbol{d},r}$ will have a unique global minimum (although there may be "spurious" critical points that arise from the non-convex parameterization).

# 3   Architecture dimensions

In this section, we begin our study of the dimension of $\mathcal{V}_{\boldsymbol{d},r}$. We describe the connection between polynomial networks and tensor decompositions for both shallow (Section 3.1) and deep (Section 3.2) networks, and we present some computational examples (Section 3.3).

## 3.1   Shallow networks and tensors

Polynomial networks with $h = 2$ are closely related to *CP tensor decomposition* [22]. Indeed in the shallow case, we can verify the network map $\Phi_{(d_0,d_1,d_2),r}$ sends $W_1 \in \mathbb{R}^{d_1 \times d_0}, W_2 \in \mathbb{R}^{d_2 \times d_1}$ to:

$$W_2 \rho_r W_1 x = \Big( \sum_{i=1}^{d_1} W_2(:,i) \otimes W_1(i,:)^{\otimes r} \Big) \cdot x^{\otimes r} =: \Phi(W_2, W_1) \cdot x^{\otimes r}.$$

Here $\Phi(W_2, W_1) \in \mathbb{R}^{d_2} \times \mathrm{Sym}_r(\mathbb{R}^{d_0})$ is a *partially symmetric* $d_2 \times d_0^{\times r}$ tensor, expressed as a sum of $d_1$ partially symmetric rank 1 terms, and $\cdot$ denotes contraction of the last $r$ indices. Thus the functional space $\mathcal{F}_{(d_0,d_1,d_2),r}$ is the set of rank $\leq d_1$ partially symmetric tensors. Algorithms for low-rank CP decomposition could be applied to $\Phi(W_2, W_1)$ to recover $W_2$ and $W_1$. In particular, when $d_2 = 1$, we obtain a symmetric $d_0^{\times r}$ tensor. For this case, we have the following.

**Lemma 8.** *A shallow architecture $\boldsymbol{d} = (d_0, d_1, 1)$ is filling for the activation degree $r$ if and only if every symmetric tensor $T \in \mathrm{Sym}_r(\mathbb{R}^{d_0})$ has rank at most $d_1$.*

Furthermore, the celebrated *Alexander-Hirschowitz Theorem* [1] from algebraic geometry provides the dimension of $\mathcal{V}_{\boldsymbol{d},r}$ for *all* shallow, single-output architectures.

**Theorem 9** (Alexander-Hirschowitz). *If $\boldsymbol{d} = (d_0, d_1, 1)$, the dimension of $\mathcal{V}_{\boldsymbol{d},r}$ is given by* $\min\left(d_0 d_1, \binom{d_0+r-1}{r}\right)$, *except for the following cases:*

- $r = 2$, $2 \leq d_1 \leq d_0 - 1$,

- $r = 3$, $d_0 = 5$, $d_1 = 7$,

- $r = 4$, $d_0 = 3$, $d_1 = 5$,

- $r = 4$, $d_0 = 4$, $d_1 = 9$,

- $r = 4$, $d_0 = 5$, $d_1 = 15$.

## 3.2   Deep networks and tensors

Deep polynomial networks also relate to a certain iterated tensor decomposition. We first note the map $\Phi_{\boldsymbol{d},r}$ may be expressed via the so-called *Khatri-Rao product* from multilinear algebra. Indeed $\theta$ maps to:

$$\mathrm{SymRow}\ W_h((W_{h-1} \ldots (W_2(W_1^{\bullet r}))^{\bullet r} \ldots)^{\bullet r}). \tag{3}$$

Here the Khatri-Rao product operates on rows: for $M \in \mathbb{R}^{a \times b}$, the power $M^{\bullet r} \in \mathbb{R}^{a \times b^r}$ replaces each row, $M(i,:)$, by its vectorized $r$-fold outer product, $\mathrm{vec}(M(i,:)^{\otimes r})$. Also in (3), SymRow denotes symmetrization of rows, regarded as points in $(\mathbb{R}^{d_0})^{\otimes r^{h-1}}$, a certain linear operator.

Another viewpoint comes from using polynomials and inspecting the layers in reverse order. Writing $[p_{\theta 1}, \ldots, p_{\theta d_{h-1}}]^T$ for the output polynomials at depth $h - 1$, the top output at depth $h$ is:

$$w_{h11}\, p_{\theta 1}^r + w_{h12}\, p_{\theta 2}^r + \ldots + w_{h1d_{h-1}}\, p_{\theta d_{h-1}}^r. \tag{4}$$

This expresses a polynomial as a weighted sum of $r$-th powers of other (nonlinear) polynomials. Recently, a study of such decompositions has been initiated in the algebra community [24]. Such expressions extend usual tensor decompositions, since weighted sums of powers of homogeneous *linear* forms correspond to CP symmetric decompositions. Accounting for earlier layers, our neural network expresses each $p_{\theta i}$ in (4) as $r$-th powers of lower-degree polynomials at depth $h - 2$, so forth. Iterating the main result in [16] on decompositions of type (4), we obtain the following bound on filling intermediate widths.

**Theorem 10** (Bound on filling widths). *Suppose $\boldsymbol{d} = (d_0, d_1, \ldots, d_h)$ and $r \geq 2$ satisfy*

$$d_{h-i} \geq \min \left( d_h \cdot r^{i(d_0-1)}, \binom{r^{h-i} + d_0 - 1}{r^{h-i}} \right)$$

*for each $i = 1, \ldots, h - 1$. Then the functional variety $\mathcal{V}_{\boldsymbol{d},r}$ is filling.*

### 3.3 Computational investigation of dimensions

We have written code[2] in the mathematical software SageMath [12] that computes the dimension of $\mathcal{V}_{\boldsymbol{d},r}$ for a general architecture $\boldsymbol{d}$ and activation degree $r$. Our approach is based on randomly selecting parameters $\theta = (W_h, \ldots, W_1)$ and computing the rank of the Jacobian of $\Phi_{\boldsymbol{d},r}(\theta)$ in (1). This method is based on the following lemma, coming from the fact that the map $\Phi_{\boldsymbol{d},r}$ is algebraic.

**Lemma 11.** *For all $\theta \in \mathbb{R}^{d_\theta}$, the rank of the Jacobian matrix $\mathrm{Jac}\,\Phi_{\boldsymbol{d},r}(\theta)$ is at most the dimension of the variety $\mathcal{V}_{\boldsymbol{d},r}$. Furthermore, there is equality for almost all $\theta$ (i.e., for a non-empty Zariski-open subset of $\mathbb{R}^{d_\theta}$).*

Thus if $\mathrm{Jac}\,\Phi_{\boldsymbol{d},r}(\theta)$ is full rank at any $\theta$, this witnesses a mathematical proof $\mathcal{V}_{\boldsymbol{d},r}$ is filling. On the other hand if the Jacobian is rank-deficient at random $\theta$, this indicates with "probability 1" that $\mathcal{V}_{\boldsymbol{d},r}$ is not filling. We have implemented two variations of this strategy, by leveraging backpropagation. Both work over a finite field $\mathbb{F} = \mathbb{Z}/p\mathbb{Z}$ to avoid floating-point computations (for almost all primes $p$, this provides the correct dimension over $\mathbb{R}$).

1. *Backpropagation over a polynomial ring.* We defined a network class over a ring $\mathbb{F}[x_1, \ldots, x_{d_0}]$, taking as input a vector variables $x = (x_1, \ldots, x_{d_0})$. Performing automatic differentiation (backpropagation) of the output function yields polynomials corresponding to $dp_\theta(x)/dw$, for any entry $w$ of a weight matrix $W_i$. Extracting the coefficients of the monomials in $x$, we recover the entries of the Jacobian of $\Phi_{\boldsymbol{d},r}(\theta)$.

2. *Backpropagation over a finite field.* We defined a network class over the finite field $\mathbb{F} = \mathbb{Z}/p\mathbb{Z}$. After performing backpropagation at a sufficient number of random sample points $x$, we can recover the entries of the Jacobian of $\Phi_{\boldsymbol{d},r}(\theta)$ by solving a linear system (this system is overdetermined, but it will have an exact solution in finite field arithmetic).

The first algorithm is simpler and does not require interpolation, but is generally slower. We present examples of some of our computations in Tables 1 and 2. Table 1 shows minimal architectures $\boldsymbol{d} = (d_0, \ldots, d_h)$ that are filling, as the depth $h$ varies. Here, "minimal" is with respect to the partial ordering comparing all widths. It is interesting to note that for deeper networks, there is not a unique

Table 1: Minimal filling widths for $r = 2$, $d_0 = 2$, $d_h = 1$

| Depth ($h$) | Degree ($r^{h-1}$) | Minimal filling ($\boldsymbol{d}$) |
|:---:|:---:|:---:|
| 3 | 4 | (2,2,2,1) |
| 4 | 8 | (2,3,3,2,1) |
| 5 | 16 | (2,3,3,3,2,1) |
| 6 | 32 | (2,3,3,4,4,2,1) |
| 7 | 64 | (2,3,4,5,6,4,2,1) |
| 8 | 128 | (2,3,4,5,7,7,6,2,1) or (2,3,5,5,7,7,5,2,1) |
| 9 | 256 | (2,3,4,8,8,8,8,8,4,1) or (2,3,4,5,8,9,9,8,4,1) |

Table 2: Examples of dimensions of $\mathcal{V}_{\boldsymbol{d},r}$

|  | $r = 2$ | $r = 3$ | $r = 4$ | $r = 5$ | $r = 6$ |
|---|---|---|---|---|---|
| $\boldsymbol{d} = (3, 2, 1)$ | 5 | 6 | 6 | 6 | 6 |
| $\boldsymbol{d} = (2, 3, 2)$ | 6 | 8 | 9 | 9 | 9 |
| $\boldsymbol{d} = (2, 3, 2, 3)$ | 10 | 12 | 13 | 13 | 13 |
| $\boldsymbol{d} = (2, 3, 2, 3, 4)$ | 16 | 21 | 22 | 22 | 22 |

minimally filling network. Also conspicuous is that minimal filling widths are "unimodal", (weakly) increasing and then (weakly) decreasing. Arguably, this pattern conforms with common wisdom.

**Conjecture 12** (Minimal filling widths are unimodal). *Fix $r$, $h$, $d_0$ and $d_h$. If $\boldsymbol{d} = (d_0, d_1, \ldots, d_h)$ is a minimal filling architecture, there is $i$ such that $d_0 \leq d_1 \leq \ldots \leq d_i$ and $d_i \geq d_{i+1} \geq \ldots \geq d_h$.*

Table 2 shows examples of computed dimensions, for varying architectures and degrees. Notice that the dimension of an architecture stabilizes as the degree $r$ increases.

## 4   General results

This section presents general results on the dimension of $\mathcal{V}_{\boldsymbol{d},r}$. We begin by pointing out symmetries in the network map $\Phi_{\boldsymbol{d},r}$, under suitable scaling and permutation.

**Lemma 13** (Multi-homogeneity). *For arbitrary invertible diagonal matrices $D_i \in \mathbb{R}^{d_i \times d_i}$ and permutation matrices $P_i \in \mathbb{Z}^{d_i \times d_i}$ ($i = 1, \ldots, h-1$), the map $\Phi_{\boldsymbol{d},r}$ returns the same output under the replacement:*

$$W_1 \leftarrow P_1 D_1 W_1$$
$$W_2 \leftarrow P_2 D_2 W_2 D_1^{-r} P_1^T$$
$$W_3 \leftarrow P_3 D_3 W_3 D_2^{-r} P_2^T$$
$$\vdots$$
$$W_h \leftarrow W_h D_{h-1}^{-r} P_{h-1}^T.$$

*Thus the dimension of a generic fiber (pre-image) of $\Phi_{\boldsymbol{d},r}$ is at least $\sum_{i=1}^{h-1} d_i$.*

Our next result deduces a general upper bound on the dimension of $\mathcal{V}_{\boldsymbol{d},r}$. Conditional on a standalone conjecture in algebra, we prove that equality in the bound is achieved for all sufficiently high activation degrees $r$. An unconditional result is achieved by varying the activation degrees per layer.

**Theorem 14** (Naive bound and equality for high activation degree). *If $\boldsymbol{d} = (d_0, \ldots, d_h)$, then*

$$\dim \mathcal{V}_{\boldsymbol{d},r} \leq \min\left(d_h + \sum_{i=1}^{h}(d_{i-1} - 1)d_i, \; d_h \binom{d_0 + r^{h-1} - 1}{r^{h-1}}\right). \tag{5}$$

*Conditional on Conjecture 16, for fixed $\boldsymbol{d}$ satisfying $d_i > 1$ ($i = 1, \ldots, h-1$), there exists $\tilde{r} = \tilde{r}(\boldsymbol{d})$ such that whenever $r > \tilde{r}$, we have an equality in (5). Unconditionally, for fixed $\boldsymbol{d}$ satisfying $d_i > 1$ ($i = 1, \ldots, h-1$), there exist infinitely many $(r_{h-1}, r_{h-2}, \ldots, r_1)$ such that the image of $(W_h, \ldots, W_1) \mapsto W_h \rho_{r_{h-1}} W_{h-1} \rho_{r_{h-2}} \ldots \rho_{r_1} W_1 x$ has dimension $d_h + \sum_i (d_{i-1} - 1)d_i$.*

**Proposition 15.** *Given positive integers $d, k, s$, there exists $\tilde{r} = \tilde{r}(d, k, s)$ with the following property. Whenever $p_1, \ldots, p_k \in \mathbb{R}[x_1, \ldots, x_d]$ are $k$ homogeneous polynomials of the same degree $s$ in $d$ variables, no two of which are linearly dependent, then $p_1^r, \ldots, p_k^r$ are linearly independent if $r > \tilde{r}$.*

**Conjecture 16.** *In the setting of Proposition 15, $\tilde{r}$ may be taken to depend only on $d$ and $k$.*

Proposition 15 and Conjecture 16 are used in induction on $h$ for the equality statements in Theorem 14. We remark that following our `arXiv` version of this paper, progress toward Conjecture 16 was made

in [30]. There, it is shown that there exists $r$ between 1 and $k!$ such that $p_1^r, \ldots, p_k^r$ are linearly independent; however, it remains open whether there exists $\tilde{r}$ as we conjecture.

The next result uses the iterative nature of neural networks to provide a recursive dimension bound.

**Proposition 17** (Recursive Bound). *For all* $(d_0, \ldots, d_k, \ldots, d_h)$ *and* $r$*, we have:*

$$\dim \mathcal{V}_{(d_0, \ldots, d_h), r} \leq \dim \mathcal{V}_{(d_0, \ldots, d_k), r} + \dim \mathcal{V}_{(d_k, \ldots, d_h), r} - d_k.$$

Using the recursive bound, we can prove an interesting *bottleneck property* for polynomial networks.

**Definition 18.** *The width $d_i$ in layer $i$ is an* asymptotic bottleneck *(for $r$, $d_0$ and $i$) if there exists $\tilde{h}$ such that for all $h > \tilde{h}$ and all $d_1, \ldots, d_{i-1}, d_{i+1}, \ldots, d_h$, then the widths $(d_0, d_1, \ldots, d_i, \ldots, d_h)$ are non-filling.*

This expresses our finding that too narrow a layer can "choke" a polynomial network, such that there is no hope of filling the ambient space, regardless of how wide elsewhere or how deep the network is.

**Theorem 19** (Bottlenecks). *If $r \geq 2, d_0 \geq 2, i \geq 1$, then $d_i = 2d_0 - 2$ is an asymptotic bottleneck. Moreover conditional on Conjecture 2 in [28], then $d_i = 2d_0$ is not an asymptotic bottleneck.*

Proposition 17 affords a simple proof that $d_i = d_0 - 1$ is an asymptotic bottleneck. However to obtain the full statement of Theorem 19, we seem to need more powerful tools from algebraic geometry.

## 5 Conclusion

We have studied the functional space of neural networks from a novel perspective. Deep polynomial networks furnish a framework for nonlinear networks, to which the powerful mathematical machinery of algebraic geometry may be applied. In this respect, we believe polynomial networks can help us access a better understanding of *deep nonlinear architectures*, for which a precise theoretical analysis has been extremely difficult to obtain. Furthermore, polynomials can be used to approximate any continuous activation function over any compact support (Stone-Weierstrass theorem). For these reasons, developing a theory of deep polynomial networks is likely to pay dividends in building understanding of general neural networks.

In this paper, we have focused our attention on the *dimension* of the functional space of polynomial networks. The dimension is the first and most basic descriptor of an algebraic variety, and in this context it provides an exact measure of the expressive power of an architecture. Our novel theoretical results include a general formula for the dimension of the architecture attained in high degree, as well as a tight lower bound and nontrivial upper bounds on the width of layers in order for the functional variety to be filling. We have also demonstrated intriguing connections with tensor and polynomial decompositions, including some which appear in very recent literature in algebraic geometry.

The tools and concepts introduced in this work for fully connected feedforward polynomial networks can be applied in principle to more general algebraic network architectures. Variations of our algebraic model could include multiple polynomial activations (rather than just single exponentiations) or more complex connectivity patterns of the network (convolutions, skip connections, *etc.*). The functional varieties of these architectures could be studied in detail and compared. Another possible research direction is a geometric study of the functional varieties, beyond the simple dimension. For example, the *degree* or the *Euclidean distance degree* [13] of these varieties could be used to bound the number of critical points of a loss function. Additionally, motivated by Section 3.2, we would like to develop computational methods for constructing a network architecture that represents an assigned polynomial mapping. Such algorithms might lead to "closed form" approaches for learning using polynomial networks (similar to SVD or tensor decomposition), as a provable counterpoint to gradient descent methods. Our research program might also shed light on the practical problem of choosing an appropriate architecture for a given application.

## Acknowledgements

We thank Justin Chen, Amit Moscovich, Claudiu Raicu and Steven Sam for helpful conversations. JK was partially supported by the Simons Collaboration on Algorithms and Geometry. MT and JB were partially supported by the Alfred P. Sloan Foundation, NSF RI-1816753 and Samsung Electronics.

## Footnotes

[1]The Zariski closure of a set $X$ is the smallest set containing $X$ that can be described by polynomial equations.

[2]Available at https://github.com/mtrager/polynomial_networks.

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
