[Supplementary Material]

## A Technical proofs

**Proposition 5** (Filling functional space). *Fix $r$ and suppose $\boldsymbol{d} = (d_0, d_1, \ldots, d_{h-1}, d_h)$ has a filling functional variety $\mathcal{V}_{\boldsymbol{d},r}$. Then the architecture $\boldsymbol{d}' = (d_0, 2d_1, \ldots, 2d_{h-1}, d_h)$ has a filling functional space, i.e., $\mathcal{F}_{\boldsymbol{d}',r} = \mathrm{Sym}_{r^{h-1}}(\mathbb{R}^{d_0})^{d_h}$.*

*Proof.* We mimic the proof of Theorem 1 in [5]. As $\mathcal{F}_{\boldsymbol{d},r}$ is thick, equivalently $\mathcal{F}_{\boldsymbol{d},r}$ contains some Euclidean open ball $B \subset \mathrm{Sym}_{r^{h-1}}(\mathbb{R}^{d_0})^{d_h}$ (see Chevalley's theorem [18]). But given any point $p \in \mathrm{Sym}_{r^{h-1}}(\mathbb{R}^{d_0})^{d_h}$, we may write $p = \lambda_1 p_1 + \lambda_2 p_2$ for some $p_1, p_2 \in B$ and $\lambda_1, \lambda_2 \in \mathbb{R}$. Thus in the architecture $\boldsymbol{d}'$, we may set the "top half" of weights to represent $p_1$, the "bottom half" to represent $p_2$, and so scaling $W_h$ appropriately, all together the network represents $\lambda_1 p_1 + \lambda_2 p_2$. $\quad\square$

**Proposition 6.** *If the closure of a set $C \subset \mathbb{R}^n$ is not convex, then there exists a convex function $f$ on $\mathbb{R}^n$ whose restriction to $C$ has arbitrarily "bad" local minima (that is, there exist local minima whose value is arbitrarily larger than that of a global minimum).*

*Proof.* We write $cl(C)$ for the closure of $C$. Let $L \subset \mathbb{R}^n$ a line that intersects $cl(C)$ in (at least) two closed disjoint intervals $L \cap cl(C) \supset I_1 \cup I_2$. Such line always exists because $cl(C)$ is not convex. It is easy to construct a convex function $f : \mathbb{R}^n \to \mathbb{R} \cup \{+\infty\}$ that is $+\infty$ outside of $L$ and has (arbitrarily) different minima when restricted to $I_1, I_2$: this amounts to constructing a convex function $\tilde{f} : \mathbb{R} \to \mathbb{R}$ with assigned minima on disjoint closed intervals. $\quad\square$

**Proposition 7.** *If a functional space $\mathcal{F}_{\boldsymbol{d},r}$ is not thick, then it is not convex.*

*Proof.* It is enough to argue that $\mathcal{F}_{\boldsymbol{d},r}$ does not lie on a linear subspace (*i.e.*, that its affine hull is the whole ambient space). Indeed, because $\mathcal{F}_{\boldsymbol{d},r}$ has zero-measure, this implies that it cannot coincide with its convex hull. To show the claim, we observe that $\mathcal{F}_{\boldsymbol{d},r}$ always contains all vectors of polynomials of the form $q_i(\ell) = [0, \ldots 0, \ell^{r^{h-1}}, 0, \ldots, 0]^T \in \mathrm{Sym}_{r^{h-1}}(\mathbb{R}^{d_0})^{d_h}$, where $\ell$ is a linear form in $d_0$ variables (this follows by induction on $h$). The vectors $q_i(\ell)$ span the whole ambient space, because any polynomial can be written as a linear combination of powers of linear forms. $\quad\square$

**Lemma 8.** *A shallow architecture $\boldsymbol{d} = (d_0, d_1, 1)$ is filling for the activation degree $r$ if and only if every symmetric tensor $T \in \mathrm{Sym}_r(\mathbb{R}^{d_0})$ has rank at most $d_1$.*

*Proof.* This is clear as the network outputs $\Phi(W_2, W_1) = \sum_{i=1}^{d_1} w_{21i} W_1(i,:)^{\otimes r} \in \mathrm{Sym}_r(\mathbb{R}^{d_0})$. $\quad\square$

**Theorem 10** (Bound on filling widths). *Suppose $\boldsymbol{d} = (d_0, d_1, \ldots, d_h)$ and $r \geq 2$ satisfy*

$$d_{h-i} \geq \min\left(d_h \cdot r^{i(d_0-1)}, \binom{r^{h-i} + d_0 - 1}{r^{h-i}}\right)$$

*for each $i = 1, \ldots, h-1$. Then the functional variety $\mathcal{V}_{\boldsymbol{d},r}$ is filling.*

*Proof.* It is equivalent to show that the network map with scalars extended to $\mathbb{C}$ (*i.e.*, allowing complex weights), denoted $\Phi_{\boldsymbol{d},r} \otimes \mathbb{C} : \mathbb{C}^{d_\theta} \to \mathrm{Sym}_{r^{h-1}}(\mathbb{C}^{d_0})^{d_h}$, has full-measure image. For this, we use induction on $h$. The key input is Theorem 4 of [16], which states generic homogeneous polynomials over $\mathbb{C}$ of degree $rs$ in $d$ variables can be written as a sum of $\leq r^{d-1}$ many $r$-th powers of degree $s$ polynomials over $\mathbb{C}$, when $r \geq 2$.

The base case $h = 1$ is trivial. Thus assume $h > 1$ and that the image has full measure for $h-1$. If $d_{h-1} \geq \binom{r^{h-1}+d_0-1}{r^{h-1}}$, then for generic $W_{h-1}, \ldots, W_1$, the entries of $\rho_r W_{h-1} \ldots \rho_r W_1 x$ form a vector space basis of $\mathrm{Sym}_{r^{h-1}}(\mathbb{C}^{d_0})$, so the image of $\Phi_{\boldsymbol{d},r} \otimes \mathbb{C}$ is filling. On the other hand if $d_{h-1} \geq d_h \cdot r^{d_0-1}$, then the image of $\Phi_{\boldsymbol{d},r} \otimes \mathbb{C}$ is full measure by [16] and the inductive hypothesis. $\quad\square$

**Lemma 11.** *For all $\theta \in \mathbb{R}^{d_\theta}$, the rank of the Jacobian matrix $\mathrm{Jac}\, \Phi_{\boldsymbol{d},r}(\theta)$ is at most the dimension of the variety $\mathcal{V}_{\boldsymbol{d},r}$. Furthermore, there is equality for almost all $\theta$ (i.e., for a non-empty Zariski-open subset of $\mathbb{R}^{d_\theta}$).*

*Proof.* We note entries of $\mathrm{Jac}\, \Phi_{\boldsymbol{d},r}(\theta)$ are polynomials in $\theta$, thus minors of $\mathrm{Jac}\, \Phi_{\boldsymbol{d},r}(\theta)$ are polynomials in $\theta$, so $\mathrm{Jac}\, \Phi_{\boldsymbol{d},r}(\theta)$ has a Zariski-generic rank (the largest size of minor that is a nonzero polynomial), which is also the maximum rank of $\mathrm{Jac}\, \Phi_{\boldsymbol{d},r}(\theta)$. By basic algebraic geometry, this is the dimension of $\mathcal{V}_{\boldsymbol{d},r}$ (see "generic submersiveness" of algebraic maps in characteristic 0 [13]). □

**Lemma 13** (Multi-homogeneity). *For arbitrary invertible diagonal matrices $D_i \in \mathbb{R}^{d_i \times d_i}$ and permutation matrices $P_i \in \mathbb{Z}^{d_i \times d_i}$ ($i = 1, \ldots, h-1$), the map $\Phi_{\boldsymbol{d},r}$ returns the same output under the replacement:*

$$W_1 \leftarrow P_1 D_1 W_1$$
$$W_2 \leftarrow P_2 D_2 W_2 D_1^{-r} P_1^T$$
$$W_3 \leftarrow P_3 D_3 W_3 D_2^{-r} P_2^T$$
$$\vdots$$
$$W_h \leftarrow W_h D_{h-1}^{-r} P_{h-1}^T.$$

*Thus the dimension of a generic fiber (pre-image) of $\Phi_{\boldsymbol{d},r}$ is at least $\sum_{i=1}^{h-1} d_i$.*

*Proof.* This is from the multi-homogeneity of the $r$-th power activation $\rho_r$ by substituting. □

**Theorem 14** (Naive bound and equality for high activation degree). *If $\boldsymbol{d} = (d_0, \ldots, d_h)$, then*

$$\dim \mathcal{V}_{\boldsymbol{d},r} \leq \min\left( d_h + \sum_{i=1}^h (d_{i-1} - 1)d_i, \; d_h \binom{d_0 + r^{h-1} - 1}{r^{h-1}} \right). \tag{5}$$

*Conditional on Conjecture 16, for fixed $\boldsymbol{d}$ satisfying $d_i > 1$ ($i = 1, \ldots, h-1$), there exists $\tilde{r} = \tilde{r}(\boldsymbol{d})$ such that whenever $r > \tilde{r}$, we have an equality in (5). Unconditionally, for fixed $\boldsymbol{d}$ satisfying $d_i > 1$ ($i = 1, \ldots, h-1$), there exist infinitely many $(r_{h-1}, r_{h-2}, \ldots, r_1)$ such that the image of $(W_h, \ldots, W_1) \mapsto W_h \rho_{r_{h-1}} W_{h-1} \rho_{r_{h-2}} \ldots \rho_1 W_1 x$ has dimension $d_h + \sum_i (d_{i-1} - 1)d_i$.*

*Proof.* We know the dimension of $\mathcal{V}_{\boldsymbol{d},r}$ equals the dimension of the domain of $\Phi_{\boldsymbol{d},r}$ minus the dimension of a generic fiber of $\Phi_{\boldsymbol{d},r}$ (see generic freeness [15]). Thus by Lemma 13, $\dim \mathcal{V}_{\boldsymbol{d},r} \leq \sum_{i=1}^h d_{i-1}d_i - \sum_{i=1}^{h-1} d_i = d_h + \sum_{i=1}^h (d_{i-1} - 1)d_i$. At the same time, the dimension of $V_{\boldsymbol{d},r}$ is at most that of its ambient space $\mathrm{Sym}_{r^{h-1}}(\mathbb{R}^{d_0})^{d_h}$. Combining produces the bound (10).

For the next statement, we temporarily assume Conjecture 16. We shall prove by induction on $h$ the stronger result that for $r \gg 0$ the generic fibers of $\Phi_{\boldsymbol{d},r}$ are precisely as described in Lemma 13 (and no more). The base case $h = 1$ is trivial. Thus assume $h > 1$ and that for $h - 1$ the generic fiber is exactly as in Lemma 13, whenever $r > \tilde{r}_1 = \tilde{r}_1(d_0, \ldots, d_{h-1})$. For the induction step, we let $\tilde{r}_2 = \tilde{r}_2(d_0, d_{h-1})$ be a threshold which works in Conjecture 16 for $d = d_0$ and $k = 2d_{h-1}$, and then we set $\tilde{r}_3 = \tilde{r}_3(d_0, \ldots, d_h) = \max(\tilde{r}_1, \tilde{r}_2)$. Now with fixed generic weights $W_h, \ldots, W_1$, we consider any other weights $\tilde{W}_h, \ldots, \tilde{W}_h$ satisfying

$$W_h \rho_r W_{h-1} \ldots \rho_r W_1 x = \tilde{W}_h \rho_r \tilde{W}_{h-1} \ldots \rho_r \tilde{W}_1 x \tag{6}$$

for $r > \tilde{r}_3$. Write $[p_{\theta 1} \quad \ldots \quad p_{\theta d_{h-1}}]$ for the output of the LHS in (6) at depth $h-1$, and similarly $[\tilde{p}_{\theta 1} \quad \ldots \quad \tilde{p}_{\theta d_{h-1}}]$ for the RHS. By genericity and $d_i > 1$, the polynomials $p_{\theta i}$ are pairwise linearly independent. Comparing the top outputs at depth $h$ in (6), we get two decompositions of type (4):

$$w_{h11} p_{\theta 1}^r + \ldots + w_{h1d_{h-1}} p_{\theta d_{h-1}}^r = \tilde{w}_{h11} \tilde{p}_{\theta 1}^r + \ldots + \tilde{w}_{h1d_{h-1}} \tilde{p}_{\theta d_{h-1}}^r. \tag{7}$$

Since $r > \tilde{r}_2$, by Conjecture 16 there must be two linearly dependent summands in (7). Permuting as necessary we may assume these are the first two terms on both sides. Scaling as necessary we may assume $p_{\theta 1} = \tilde{p}_{\theta 1}$, and then subtract $\tilde{w}_{h11} \tilde{p}_{\theta 1}^r$ from (7) to get:

$$(w_{h11} - \tilde{w}_{h11}) p_{\theta 1}^r + \ldots + w_{h1d_{h-1}} p_{\theta d_{h-1}}^r = \tilde{w}_{h12} \tilde{p}_{\theta 2}^r + \ldots + \tilde{w}_{h1d_{h-1}} \tilde{p}_{\theta d_{h-1}}^r. \tag{8}$$

Invoking Conjecture 16 again, we may remove another summand from the RHS, so on until the RHS is 0. Then each individual summand in the LHS must be 0 too, by pairwise linear independence and Conjecture 16 once more. We have argued that (up to scales and permutation) it must hold $[p_{\theta 1} \ \ldots \ p_{\theta d_{h-1}}] = [\tilde{p}_{\theta 1} \ \ldots \ \tilde{p}_{\theta d_{h-1}}]$ and $W_h(1,:) = \tilde{W}_h(1,:)$. Comparing other outputs at depth $h$ in (6) gives $W_h = \tilde{W}_h$ (up to scales and permutation). Thus by the inductive hypothesis, the fiber through $(W_h, \ldots, W_1)$ is as in Lemma 13 and no more. This completes the induction.

For the unconditional result with differing degrees per layer, the argument runs closely along similar lines, but it relies on Proposition 15 in place of Conjecture 16. For brevity, the details are omitted. $\square$

**Proposition 15.** *Given positive integers $d, k, s$, there exists $\tilde{r} = \tilde{r}(d, k, s)$ with the following property. Whenever $p_1, \ldots, p_k \in \mathbb{R}[x_1, \ldots, x_d]$ are $k$ homogeneous polynomials of the same degree $s$ in $d$ variables, no two of which are linearly dependent, then $p_1^r, \ldots, p_k^r$ are linearly independent if $r > \tilde{r}$.*

*Proof.* It is shown in [4] (via Wronskian and Vandermonde determinants) that for any *particular* $p_1, \ldots, p_k$, no two of which are linearly dependent, there exists $\tilde{r} = \tilde{r}(p_1, \ldots, p_k)$ such that $p_1^r, \ldots, p_k^r$ if $r > \tilde{r}$. The dependence on particular $p_1, \ldots, p_k$ can be removed as follows.

Let $U \subset \operatorname{Sym}_s(\mathbb{R}^d)^k$ be the set of $k$-tuples, no two entries of which are linearly dependent. So $U$ is Zariski-open, described by the non-vanishing of $2 \times 2$ minors. Further let $U_r \subseteq U$ be the subset of $k$-tuples whose $r$-th powers are linearly independent, similarly Zariski-open. Consider the chain of inclusions $U_1 \subseteq U_1 \cup U_2 \subseteq U_1 \cup U_2 \cup U_3 \subseteq \ldots$. By [4], the union of this chain equals $U$. Thus by Noetherianity of affine varieties, there exists $R$ with $\cup_{r=1}^R U_r = U$ [15]. Now $\tilde{r} = R!$ works. $\square$

**Proposition 17** (Recursive Bound). *For all $(d_0, \ldots, d_k, \ldots, d_h)$ and $r$, we have:*
$$\dim \mathcal{V}_{(d_0, \ldots, d_h), r} \leq \dim \mathcal{V}_{(d_0, \ldots, d_k), r} + \dim \mathcal{V}_{(d_k, \ldots, d_h), r} - d_k.$$

*Proof.* This bound encapsulates the bracketing:
$$(W_h \rho_r W_{h-1} \ldots W_{k+1}) \rho_r (W_k \rho_r W_{k-1} \ldots W_1 x). \tag{9}$$
More formally, the network map $\Phi_{(d_0, \ldots, d_h), r}$ factors as:
$$\mathbb{R}^{d_\theta} \longrightarrow \operatorname{Sym}_{r^{h-k-1}}(\mathbb{R}^{d_k})^{d_h} \times \operatorname{Sym}_{r^{k-1}}(\mathbb{R}^{d_0})^{d_k} \longrightarrow \operatorname{Sym}_{r^{h-1}}(\mathbb{R}^{d_0})^{d_h} \tag{10}$$
by first sending $(W_h, \ldots, W_1)$ to the pair of bracketed terms in (9) and then the pair to the composite in (9). The closure of the image of the first map in (10) is $\mathcal{V}_{(d_0, \ldots, d_h), r} \times \mathcal{V}_{(d_k, \ldots, d_h), r}$. On the other hand, the second map in (10) has $\geq d_k$-dimensional generic fibers, by multiplying with a diagonal matrix $D_k \in \mathbb{R}^{d_k \times d_k}$. Combining these facts gives the result. $\square$

**Theorem 19** (Bottlenecks). *If $r \geq 2, d_0 \geq 2, i \geq 1$, then $d_i = 2d_0 - 2$ is an asymptotic bottleneck. Moreover conditional on Conjecture 2 in [28], then $d_i = 2d_0$ is not an asymptotic bottleneck.*

*Proof.* We first point out that Proposition 17 gives an elementary proof $d_i = d_0 - 1$ is an asymptotic bottleneck. This is because as $h$ grows the ambient dimension grows like $O(d_h \cdot d_0^{r^{h-1}})$, while the RHS bound grows like $O(d_h \cdot d_i^{r^{h-i-1}})$, so if $d_i < d_0$ then $\mathcal{V}_{d,r}$ cannot fill for $h \gg 0$.

To gain a factor of 2 in the bottleneck bound, we start by writing $[p_{\theta 1} \ \ldots \ p_{\theta d_i}]^T$ for the output polynomials at depth $i$, that is, for $W_i \rho_r W_{i-1} \ldots \rho_r W_1 x$. Fixing $\theta$, we consider $A_\theta := \mathbb{R}[p_{\theta 1}^r, \ldots, p_{\theta d_i}^r]$, a subalgebra of the *Veronese ring* $V_{d_0, r^i} := \mathbb{R}[x_1^{\alpha_1} \ldots x_{d_0}^{\alpha_{d_0}} : \sum_{j=1}^{d_0} \alpha_j = r^i]$. The key idea is to compare the *Hilbert polynomials* of $A_\theta$ and of $V_{d_0, r^i}$ [6]. If the Hilbert polynomials *differ in any non-constant terms*, this means the dimension of the degree $D$ piece of $A_\theta$ minus that of $V_{d_0, r^i}$ diverges to $-\infty$ as $D$ goes to $\infty$. At the same time, however we vary weights $W_{i+1}, \ldots, W_h$ (keeping $\theta = W_1, \ldots, W_i$ fixed), the output polynomials $\Phi_{d,r}$ remain in the algebra $A_\theta$. Additionally, for varying $\theta$ and $d_1, \ldots, d_{i-1}$, the possible $d_i$-vectors of degree $r^i$ polynomials in $d_0$ variables, $[p_{\theta 1} \ \ldots \ p_{\theta d_i}]^T$, comprise a bounded-dimensional variety. The upshot is that if it need always be the case (based on $r, d_0, i, d_i$) that the Hilbert polynomials of $A_\theta$ and $V_{d_0, r^i}$ have non-constant difference, then $d_i$ must be an asymptotic bottleneck. Thus it suffices to check the Hilbert polynomial property holds for all $\theta$ if $d_i = 2d_0 - 2$. To this end, we derived the following general result:

**Claim.** *Given integers $d \geq 2$ and $s \geq 2$. Then whenever $p_1, \ldots, p_{2d-2} \in \mathbb{R}[x_1, \ldots, x_d]$ are $2d-2$ homogeneous polynomials of the same degree $s$ in $d$ variables, the algebra $\mathbb{R}[p_1, \ldots, p_{2d-2}]$ and the Veronese algebra $V_{d,s}$ have Hilbert polynomials with non-constant difference.*

*Proof of claim.* First, it suffices to check the claim for generic $p_i$. Second, the difference in Hilbert polynomials identifies with the Hilbert polynomial of the *sheaf* $\mathcal{G} = \mathrm{coker}(\mathcal{O}_Y \to \pi_* \mathcal{O}_X)$ [18]. Here $X := \mathbb{V}_{d,s} \subset \mathbb{P}^{N_{s,d}-1}$ ($N_{s,d} = \binom{d+s-1}{s}$) is the *projective Veronese variety*, the linear projection $\pi : \mathbb{P}^{N_{s,d}-1} \dashrightarrow \mathbb{P}^{2d-3}$ corresponds to $(p_1, \ldots, p_{2d-2})$, and finally $Y := \overline{\pi(X)}$ is the closure of $X$ projected by $\pi$. By general facts, the degree of the Hilbert polynomial of $\mathcal{G}$ equals the projective dimension of the support of $\mathcal{G}$, and this support is the *branch locus* of $\pi|_X$. Now let $L \subset \mathbb{P}^{N_{s,d}-1}$ denote the base locus (kernel) of $\pi$, a linear subspace of projective dimension $N_{s,d} - 2d + 1$. If $d \geq 3$, $s \geq 3$, then $L \cap Sec(X)$ is a curve, where $Sec$ denotes the line *secant variety* [22] ($d = 2$ or $s = 2$ are omitted simple special cases). Each point on $L \cap Sec(X)$ lies on a line through two points on $X$; these points map to the same image under $\pi$, giving a point in the branch locus of $\pi|_X$. It follows the branch locus is a curve, thus the degree of the Hilbert polynomial of $\mathcal{G}$ is $1 > 0$, as desired. $\quad\square$

By the preceding discussion, the claim establishes $d_i = 2d_0 - 2$ is an asymptotic bottleneck.

For the statement when $d_i = 2d_0$, let us temporarily assume Conjecture 2 in [28]. This means $\mathbb{R}[p_{\theta 1}^r, \ldots, p_{\theta 2d_0}^r]$ has the same Hilbert function as $\mathbb{R}[P_1, \ldots, P_{2d_0}]$ for generic forms $P_i$ of degree $r^i$, provided $p_{\theta i}$ are generic forms of degree $r^{i-1}$. Reasoning as for the claim, $\mathbb{R}[P_1, \ldots, P_{2d_0}]$ has the same Hilbert polynomial as the Veronese ring $V_{d_0, r^i}$. Thus if we choose $(d_1, \ldots, d_{i-1})$ so that $(d_0, \ldots, d_i)$ is filling, then it follows we can choose $h \gg 0$ and $(d_{i+1}, \ldots, d_h)$ so that $(d_0, \ldots, d_h)$ is filling. In other words, $d_i = 2d_0$ is not an asymptotic bottleneck. $\quad\square$