[Reviews · NeurIPS 2019]

Reviewer 1



Post-rebuttal: After reading the authors' response and further consideration, I am downgrading my score to 7 from 9. While I am still very excited about the new perspective this work brings, I now realize that there is still a lot of work remaining in order to tie the theoretical results to real-world phenomena. Regardless of whether the paper gets accepted, I'd ask the authors to make the gap clearer and to lay out more clearly an agenda for future work that address the various issues discussed in the rebuttal, e.g.: approximation, empirical notions of filling, etc. =============== ORIGINALITY =========== The paper considers the functional space of polynomial networks as an algebraic object. They use tools from algebraic geometry to analyze the dimension of the Zariski closure of this space. This perspective greatly generalizes previous work on two-layer networks with quadratic activations. The paper is highly original in relating recent results from algebra to basic issues about neural networks. QUALITY & CLARITY ================= This work tackles head-on the problem of analyzing the functional space of polynomial varieties. It gives a clear overview of the model and an understandable interpretation of the technical results. The proofs deferred to the appendix use some high-powered recent results from algebra, but the main text does well in giving a high-level overview and extracts meaningful results such as Theorem 19 about bottlenecks. I did not have the background to verify all the math but to the extent that the quoted results in the cited papers apply, the main bounds claimed in the theorems are correct. The work is clearly of high technical quality. SIGNIFICANCE ============ This paper offers new insights how the architecture of a network corresponds to its expressive power. It substantiates common wisdom about the widths of the network being unimodal for expressive networks. It also shows that if the width becomes too small at any level, it can choke the expressivity of the network. It is highly significant to see rigorous justifications of these intuitions. The only drawback to the work is that it only considers exact expression and not approximation. Is there a way to talk about approximation using this language, perhaps using something like Proposition 5 to relate it to the exact case?

Reviewer 2



==== Summary ==== This paper studies deep neural networks with polynomial activations by mapping the parameters of a fixed network's architecture to its polynomial coefficients, whose image corresponds to an algebraic variety. This connection motivates studying the fundamental properties of these varieties, namely, the dimension of the variety and whether it is filling the functional space. The paper relates the dimension to the expressivity of the network, and the filling property is shown to be helpful for optimization. The article follows by bounding the dimension for various settings and giving sufficient conditions for the variety to be filling. ==== Detailed Review ==== The connection this paper makes between neural networks and algebraic varieties seems like a fascinating and promising direction for studying neural networks. However, it is a bit difficult to see how the current results lead to a better understanding of the expressiveness and optimization neural networks. 1. While the dimension of the variety is clearly linked in some sense to the expressiveness of the architecture, it is difficult to see how this measure translates to actual measures of interest to expressiveness analysis. The fact that the analysis depends on exact equality means that we cannot ask important questions such as how many hidden units are needed to approximate the functions computed by deeper networks, or ones with a higher degree activation functions. The only clue provided by the authors is that the dimension is an upper bound on the number of examples a network can interpolate, i.e., memorize. However, the naive dimension bound provided matches exactly with other works on memorization, i.e., it is proportional to the number of parameters in a network. 2. In terms of the filling widths, this seems more readily applicable. However, it appears that even for very small input spaces (e.g., 28x28 MNIST images) and squared activations the required minimal widths are already infeasible (e.g. for d0 = 768, d_h = 1, h = 3, d_2 will have to be at least 1e10). So, while the definition seems like it could be relevant, the sufficient conditions will not be met for most of the common architectures. On a more minor note/question, regarding the computational estimation of the ranks: are the ranks computed to give exact answers (e.g., by an exact computation of the determinant of an integer matrix), or are you using the standard floating-point numerical methods that are only estimates? I suggest the authors emphasize this aspect in the paper (in a footnote or appendix), because it is unclear if table 1 and 2 are just for intuition, or actual proved values (using an exact rank method). All of the above is not to detract from the foundations laid down by the authors to what seems like a fascinating direction for analyzing neural networks. This seems like an excellent start at what could in the future yield actual results that are relevant. The paper is also clearly written, and though I was not very familiar with the underlying math of algebraic varieties, it was still easy to follow and enjoyable to read. It is for this reason that I am marginally in favor of accepting this paper as-is because it could spark further ideas and discussion that might progress our understanding.

Reviewer 3



It would have been nicer to see more investigation of whether such networks can be trained to learn polynomials of a certain degree -- something more than just what is stated in section 2.3 Another point is that there may be advantages to use activations like sigmoid that have an infinite taylor series as they can be used to approximate polynomial of any degree with one hidden layer. So its well known that a polynomial of degree d in n variables can be *learnt* (not just represented ) using about n^O(d) hidden nodes in a 2 layer network with certain activations that have infinite taylor series. The writing style keeps the reader interested but it would be better to list the main results upfront somewhere near the introduction. Its harder to get the main results now as many Theorem statements are embedded in later sections.

[Author Response · NeurIPS 2019]

# On the Expressive Power of Deep Polynomial Neural Networks

We thank the reviewers for their positive and useful comments. One shared concern among the reviewers seems to be that our study of the *exact* functional space and its dimension might not be directly helpful for understanding the practical ability of a network to represent functions *approximately*. This criticism may be prompted in part by our claim that "we do not emphasize approximation properties, but rather the study of the functions that can be expressed exactly using a network" (Sec.1.1). While it is true that we do not directly address the issue of approximation, our theory also suggests a geometric framework for characterizing the class of functions that can be approximated well by a neural network. In particular, when the functional variety is not filling, one can naturally consider functions that are "close" to the variety (nearness could be formalized by considering tubular neighborhoods of the variety). Given that our functional varieties are generalizations of families of low-rank tensors, this perspective might lead to quantitative results in terms of generalized versions of SVD (applied for example to tensor flattenings). In fact, motivated by the reviewers' questions, we plan to think about computational methods for "projecting" a polynomial on a functional variety. In a broader sense, we believe that a theory of "exact expressivity" of neural networks should have greater explanatory power compared to a purely approximation-based analysis. The fact that the former perspective has received less attention could be due to the lack of appropriate tools for addressing it; we hope that our algebraic framework can open the door to new work on this topic. If the paper is accepted, we will mention these directions, and remove the aforementioned sentence from Sec.1.1.

**Reviewer 1.**

- *"I'd ask the authors to respond to my above question about approximation."* See our detailed answer above.

**Reviewer 2.**

- *"it is difficult to see how this measure translates to actual measures of interest to expressiveness analysis [...] the naive dimension bound provided matches exactly with other works on memorization, i.e., it is proportional to the number of parameters in a network."* In addition to our general answer above, we believe that the algebraic framework may be used to derive new approximation bounds. For example, while it is true that our naive dimension bound is proportional to the number of parameters, the existence of "asymptotic bottlenecks" (Theorem 19) shows that in many situations these two quantities are very different (in the presence of a bottleneck, even if widths grow arbitrarily, the functional dimension stays bounded): our theory detects this discrepancy, and the corresponding effect on expressivity.

- *"it appears that even for very small input spaces (e.g., 28x28 MNIST images) and squared activations the required minimal widths are already infeasible"* We agree that the filling conditions are unlikely to be satisfied in practice, and for this reason we believe that learning in real-life architectures takes place in a non-filling regime. Still, qualitative distinctions between filling and non-filling architectures are theoretically important, and consistent with numerous existing results showing that learning is easier in the infinite-width setting. Furthermore, we believe that refined notions of filling will help bring our theory closer to practice. Specifically, one could consider "empirical filling", *i.e.*, whether given any sample set of a fixed size, there exist weights that perfectly interpolate the data. Also, a notion of "relative filling" could help compare different architectures, *i.e.*, when two architectures give the same functional space. We will remark in the paper how these notions could be studied in future work.

- *"are the ranks computed to give exact answers [...], or are you using the standard floating-point numerical methods that are only estimates? I suggest the authors emphasize this aspect in the paper."* Our computations are based on finite field arithmetic, using a large prime number to define the base field. As we will clarify in the paper, all filling dimensions are provably correct over $\mathbb{R}$ while all other computed dimensions are correct over $\mathbb{R}$ with very high probability.

**Reviewer 3.**

- *"It would have been nicer to see more investigation of whether such networks can be trained to learn polynomials"* The focus of this paper was on expressivity, and we only briefly touched upon optimization/learning in Sec.2.3. We are currently working on developing the connection between filling architectures and optimization further. For example, we conjecture that the absence of non-global local minima in the landscape of deep polynomial networks is actually equivalent to the condition that the functional space is filling. If true, this property would vastly generalize known results on shallow quadratic networks. We are also investigating how refined notions of filling (see above) may yield favorable optimization properties. We believe however that these issues fall outside the scope of the current paper.

- *"Another point is that there may be advantages to use activations like sigmoid"* In general, we agree that there may be benefits in using non-polynomial activations, as universal approximation theorems require them. However, we are unsure about the practical difference between using polynomial activations and ReLU or sigmoids (folklore seems to say that the choice of the activation is not so important). For example, there are optimization methods that involve expressing a non-smooth function (*e.g.*, absolute value) as a limit of polynomial functions ("$\eta$-trick"). Spelling out the connection between polynomial networks and sigmoid or ReLU networks is an important issue that we will look into.

- *"it would be better to list the main results upfront"* We thank the reviewer for this useful suggestion. Unfortunately, for most of our results, a precise formulation of the statement requires notions and terminology that are only introduced in Sec.2. However, if the paper is accepted, we will include informal versions of our main results at the end of Sec.1.

[Meta-Review · NeurIPS 2019]

The reviewers all agree that this paper has the potential to open up exciting new directions of future work. In order to have most impact, the authors should very carefully take into account the reviewer comments.